# Bi-Directional, Day-to-Day Associations between Objectively-Measured Physical Activity, Sedentary Behavior, and Sleep among Office Workers

**DOI:** 10.3390/ijerph18157999

**Published:** 2021-07-28

**Authors:** Emerald G. Heiland, Örjan Ekblom, Emil Bojsen-Møller, Lisa-Marie Larisch, Victoria Blom, Maria M. Ekblom

**Affiliations:** Department of Physical Activity and Health, The Swedish School of Sport and Health Sciences (GIH), Lidingövägen 1, 114 86 Stockholm, Sweden; orjan.ekblom@gih.se (Ö.E.); emil.bojsen.moller@gih.se (E.B.-M.); lisa-marie.larisch@gih.se (L.-M.L.); victoria.blom@gih.se (V.B.); maria.ekblom@gih.se (M.M.E.)

**Keywords:** sleep, sedentary, MVPA, office workers, physical activity, accelerometry, actigraphy

## Abstract

The bi-directional, day-to-day associations between daytime physical activity and sedentary behavior, and nocturnal sleep, in office workers are unknown. This study investigated these associations and whether they varied by weekday or weekend day. Among 324 Swedish office workers (mean age 42.4 years; 33.3% men), moderate-to-vigorous physical activity (MVPA), and sedentary behaviors and sleep (total sleep time (TST) and sleep efficiency (SE)) were ascertained by using accelerometers (Actigraph GT3X) over 8 days. Multilevel linear mixed models were used to assess the bi-directional, day-to-day, within-person associations. Additional analyses stratified by weekend/weekday were performed. On average, participants spent 6% (57 min) of their day in MVPA and 59% (9.5 h) sedentary, and during the night, TST was 7 h, and SE was 91%. More daytime sedentary behavior was associated with less TST that night, and reciprocally, more TST at night was associated with less sedentary behavior on the following weekday. Greater TST during the night was also associated with less MVPA the next day, only on weekdays. However, daytime MVPA was not associated with TST that night. Higher nighttime SE was associated with greater time spent sedentary and in MVPA on the following day, regardless if weekday or weekend day. Sleep may be more crucial for being physically active the following day than vice versa, especially on weekdays. Nevertheless, sedentary behavior’s relation with sleep time may be bi-directional. Office workers may struggle with balancing sleep and physical activity time.

## 1. Introduction

Adequate sleep duration of good quality can have positive effects on mental health, mood, alertness, cognitive performance, and productivity [1,2]. Therefore, recent interest has been directed towards improving sleep as a strategy to benefit mental health and well-being. In turn, this may be advantageous for workplace performance. However, most previous studies have used self-reported measures to assess sleep, which is colored by systematic over-reporting [3]. Therefore, more studies using device-based measurement tools are warranted.

Another modifiable lifestyle factor interrelated with sleep is physical activity. Previously, the relationship between these two factors has been described such that improving one behavior improves the other [4]. Earlier studies have, nonetheless, predominantly examined the between-person associations, used self-reporting or less ecologically valid methods (e.g., electroencephalogram or polysomnography in a laboratory), and focused only on one direction of the association, namely the effect of physical activity on sleep [4]. Studies examining the day-to-day, within-person, and bi-directional relationship between physical activity and sleep in natural environments, using device-based methods (e.g., accelerometers), are lacking in working-age populations, with only one previous study performed in women working in the US. This study found no day-to-day reciprocal associations between total sleep time (TST) or sleep efficiency (SE, a measure of sleep quality) and moderate-to-vigorous physical activity (MVPA) [5]. This study also reported the association between sedentary behavior and sleep, finding no associations [5]. This demonstrates the large gap in the literature on the working-age population. Interestingly, an experimental study has shown that sleep restriction can lead to a reduction in physical activity behavior on the follow day [6], possibly due to acute physiological changes and increased fatigue [7]; however, whether this also occurs in a free-living setting among office workers remains unknown.

Furthermore, extensive sedentary behavior is generating more interest in research owing to its link with many negative health outcomes, such as increased risk of psychological illness, cardiovascular and cerebrovascular diseases, and early mortality [8,9,10]. These associations are believed, to some extent, to be independent of physical activity’s positive effects [11]. Nevertheless, little attention has been given to sedentary and poor sleep, despite their assumed interrelatedness [5,12] and their shared negative effects on health and survival [13,14]. Specific focus on office workers in studies is needed, as this population is among the most sedentary [15], with an estimated 65–75% of their working hours spent sitting [16].

A better understanding into the interrelatedness of physical activity, sedentary behavior, and sleep is necessary to inform interventions that target any of these behaviors. Even more importantly, if there is a lack of association, this should be communicated so that focus in interventions can be placed on true correlates. Furthermore, certain socio-demographic factors, such as sex, age, and cardiorespiratory fitness [4,17], have been found as potential effect modifiers in the association between physical activity and sleep [4]. However, the role of cognitive performance in the association between physical activity and sleep remains obscure. A previous study of office workers found that average MVPA was not associated with cognitive performance, whereas sedentary behavior was [18]. However, it has also been reported that acute physical activity may enhance cognitive performance [19]. Short sleep duration or poor sleep quality, on the other hand, may impair cognitive performance [20]. Moreover, physical activity, sedentary, and sleep behaviors are likely to vary between weekday and weekend day [21], especially among office workers, but this remains unclear.

Therefore, we aimed to investigate whether (1) time spent in daytime MVPA and sedentary behavior were associated with that night’s TST and SE and (2) if nocturnal TST and SE were associated with MVPA and sedentary behavior the following day. In addition, the roles of age, sex, fitness, and cognitive performance were assessed based on these associations and whether the associations varied by weekday or weekend day.

## 2. Materials and Methods

### 2.1. Study Design and Participants

This study is part of a larger project, “Physical Activity and Healthy Brain Functions”, performed at The Swedish School of Sport and Health Sciences (GIH), in Stockholm, Sweden, and was carried out 2016–2017 [18]. Participants were recruited via mail from two companies in Stockholm and Gothenburg to take part in a cross-sectional study, through convenience sampling, and were required to have an office-based job at these companies. Initially, 1940 persons were invited to participate, of which 547 (28%) answered the web questionnaire. About two weeks later, these employees were invited to a test session where they were equipped with an activity monitor by a test leader and received a diary to record times they went to bed and woke up. There were 369 (19%) employees who attended the test session, participating in the submaximal cycle ergometer and cognitive testing, and provided activity monitor and diary data. The final analytical sample included 324 employees with complete physical activity and sleep data. Reasons for not attending the test-session were related to not having a normal working week, incomplete data, and not returning the activity monitor on time. Compared with the analytical sample, those who were missing physical activity and sleep data did not significantly differ in demographic characteristics, except that they were more likely to have poorer performance on the executive function cognitive tests.

### 2.2. Accelerometer Data Acquisition

Physical activity was measured for eight days using the Actigraph GT3X (Actigraph, Pensacola, Florida, USA), attached to the hip with an elastic band during all waking hours of the day with the exception of water-based activities. During the night, the Actigraph was placed around the non-dominant wrist.

Physical activity and sleep data were sampled at a frequency of 30 Hz [22]. Data extraction was performed using 60 s epochs with a low frequency extension filter using the ActiLife software (Actigraph, Pensacola, Florida, USA) [22].

### 2.3. Physical Activity Variables

Non-wear time was defined as 60 min or longer with zero counts. Total wear time was calculated by first subtracting time in bed (reported in the diaries), from total time, to retrieve wake time, and then subtracting non-wear time from wake time. Participants with at least four days of wear time, with a minimum of 600 min wear time per day, were included in the analysis [22].

Data were downloaded from the ActiLife software as counts per minute, and each minute was classified based on its intensity. A cut-off of >2690 counts per minute was used to classify MVPA [23], and 0–199 counts per minute was defined as sedentary behavior [24]. The proportion of each day spent in MVPA and in sedentary behavior was calculated by dividing the time spent in the respective behavior by the total wear time of that day and presented as percentages.

### 2.4. Sleep Variables

Actigraph GT3X was used to estimate sleep [25]. Participants moved the device from the hip to their non-dominant wrist when going to bed. Sleep data were processed in ActiLife using the Cole-Kripke algorithm [26]. Total sleep time (TST) in minutes and sleep efficiency as a percentage (SE = total sleep time/total minutes in bed × 100) [27] were acquired from the algorithm. Nights with no signal for at least two consecutive hours were excluded from the analysis, as well as nights without information about in-bed time in the sleep diaries. Participants with at least four valid nights were included in the analyses.

### 2.5. Other Variables

Other variables used as confounders and modifiers included age, sex, cardiorespiratory fitness, cognitive performance, and weekday vs. weekend day. Cardiorespiratory fitness was assessed using the submaximal Ekblom-Bak cycle ergometer test [28,29] performed on a Monark 838E (Monark, Varberg, Sweden) as described in Pantzar et al. [30]. A composite score of three executive function tests was used to assess cognition. The composite included the Stroop test, the Trail Making Test B (TMT-B), and the two-back test [30]. The outcome score for Stroop and TMT-B was completion time, and for the two-back it was the number of correct answers. The two-back grading scale was inverted, and the three tests were each standardized. The scores were subsequently added together and divided by the total number of tests in order to create a composite score. Stress was assessed as a single-item measure on a 5-point Likert scale [31], where the participant was asked “Do you feel this kind of stress these days? Stress is here defined as a situation in which a person feels tense, restless, nervous or anxious, or is unable to sleep at night because their mind is troubled all the time”. The weekend variable was defined as the outcome taking place on a weekend day vs. a weekday.

### 2.6. Statistical Analysis

Differences between men and women in descriptive variables were performed using Student’s t-tests for continuous parametric variables. Linear mixed models with sex as a fixed effect and subject as a random effect were performed to determine whether the within-person effects of physical activity, sedentary behavior, and sleep differed between men and women across weekdays, weekend days, and all days. In order to examine within-person effects between physical activity, sedentary behavior, and sleep, multilevel modeling for repeated measures were employed [32] in a similar manner as reported in Dzierzewski et al. [33]. All the multilevel linear mixed models (MLMs) were estimated using restricted maximum likelihood with an unstructured variance-covariance structure. All dependent and main independent variables were standardized into z-scores prior to estimation of the MLM to facilitate interpretation of parameters. MVPA and SE were skewed and therefore log-transformed before standardization. Two sets of MLMs were run to study the intra-individual differences.

The first set of MLMs was parametrized to test whether day-to-day, daytime MVPA and sedentary behavior (independent variables) were associated with that night’s TST and SE (dependent variables). MVPA or sedentary behavior was introduced in the MLMs as person-centered daily variations of these behaviors as both fixed and random effects (i.e., within-person random effects computed as the variation of time spent in either MVPA or sedentary behavior around the weekly averages). Averages of these behaviors were also controlled for in the models. Equations describing each model can be found in the Appendix A.

The second set of MLMs was used to test whether sleep parameters (TST and SE) were associated with MVPA or sedentary behavior the following day. The dataset was shifted so that physical activity/sedentary behavior the next day matched sleep the night before in the same row. MVPA and sedentary behavior were the dependent variables, and sleep parameters from the previous night were independent variables, included as person-centered, within-person, fixed and random effects.

In model 1, MVPA and sedentary behavior were mutually controlled for. In model 2, subject-level covariates (i.e., age, sex, cognitive performance, fitness) were additionally controlled for. Interactions were tested between the main independent variables and potential modifiers (i.e., age, sex, fitness, cognitive performance, and weekend/weekday). Weekend/weekday was used as a binary dummy variable in the models. Subsequent stratified MLMs were performed for statistically significant interactions (*p* < 0.1).

Sensitivity analyses, taking into consideration education and self-perceived stress as potential confounders, were performed, although there were many missing in the stress variable (*n* = 22). Additionally, the models were rerun using logistic mixed-effects models where SE and MVPA were set as outcome variables, to test whether the results differed when the respective outcomes were dichotomized based on their median splits. Statistical significance was set at a *p*-value <0.05. Statistical software Stata version 15 (StataCorp, College Station, TX, USA) was used for all analyses.

## 3. Results

The average age of the 324 participants was 42.4 years (range 24–66 years), 33.3% were men, and the average years of education was 14.4. In the analytical sample, education, body mass, fitness, and two of the cognitive tests were significantly different between women and men (Table 1). The participants in this study slept on average 416 min (equivalent to 6.9 h) per night with a median SE of 91.1%. The average awake wear time spent in sedentary behavior was 59.2% (9.5 h), and the median time for MVPA was 6.0% (57 min) (Table 2). Over all days, women significantly spent less time in sedentary behavior but more time in MVPA and TST compared with men. When stratified by weekend/weekday, sedentary time and TST differed for men and women on both weekdays and weekend days, whereas MVPA differed between men and women only on weekdays. SE did not differ between men and women.

### 3.1. Daytime Physical Activity and Sedentary Behavior’s Association with That Night’s Sleep

Proportion of awake wear time spent in sedentary behavior during the day was significantly negatively associated with TST that night. This association remained after controlling for MVPA and other confounders, suggesting that above-average time spent sedentary during the day was associated with below-average TST during the night (Table 3). Daytime MVPA was not associated with TST that night. Neither were sedentary behavior or MVPA associated with SE during the night.

### 3.2. Sleep’s Association with Physical Activity and Sedentary Behavior on the Following Day

TST was associated with sedentary time and MVPA, such that above-average TST during the night was associated with below-average sedentary time and MVPA on the following day in the fully-adjusted models (Table 3). Furthermore, above-average night’s SE was significantly associated with above-average sedentary behavior and above-average MVPA during the following day.

### 3.3. Modifiers

There were no modifying effects of fitness on any of the associations. However, age modified the association between daytime sedentary behavior and that night’s TST, such that the negative association was significant only among the office workers in the upper age range (age 37–66) (ages 21–36 years: adjusted standardized β = −0.08, 95% CI −0.16, 0.01; ages 37–66 years: β −0.15, 95% CI −0.20, −0.10).

Sex modified the association between MVPA and that night’s SE, demonstrating that above-average time spent in MVPA was associated with below-average SE during the subsequent night, but only in men (adjusted standardized β for men: −0.07, 95% CI −0.13, −0.01; women: β 0.03, 95% CI −0.01, 0.07).

Table 4 displays the stratified results by the outcome occurring either on a weekday or weekend day. Weekend/weekday showed a significant interaction with MVPA on the association with that night’s TST, and in the opposite temporal direction of TST on MVPA. More MVPA during the day was associated with below average TST during the night only for weekdays, but not weekends. However, this relation became non-significant after adjusting for age, sex, fitness, and cognitive performance. An above average TST during the night was associated with below average MVPA the next day only for weekdays not weekend days in the fully-adjusted model. Weekend/weekday also modified the association between TST during the night and sedentary behavior on the following day, such that above average TST during the night was associated with below average sedentary behavior on the following day, if a weekday, but not on a weekend day.

### 3.4. Sensitivity Analyses

When self-perceived stress was controlled for in the models, the results did not differ from the main analysis, except that the association between nighttime’s SE and MVPA on the following day was no longer significant (Appendix A, Model 1). The results did not differ when education was additionally controlled for (Appendix A, Model 2). In addition, using the logistic mixed-effects models with SE and MVPA as dichotomized outcome variables did not alter the associations demonstrated in the original analysis (Appendix A).

## 4. Discussion

The aim of this study was to investigate the day-to-day, within-person associations between accelerometer-measured proportions of MVPA and sedentary time during the day on sleep parameters, TST and SE, during the night, and in the reciprocal, temporal direction—nocturnal sleep on MVPA and sedentary time the following day. In this population of office workers, sleep was observed to be associated with MVPA on the following day; however, daytime MVPA was not associated with that night’s sleep. Moreover, the associations between sleep and sedentary behavior were significant in both temporal directions. These associations differed, however, by weekend or weekday, such that sleep was significantly associated with sedentary behavior and MVPA on the following day if the physical activity or sedentary behavior took place on a weekday rather than a weekend day. Furthermore, some of the associations differed by age and sex, whereas fitness and cognitive performance did not modify or confound any of the associations.

Previous studies examining the bi-directional, day-to-day associations between physical activity and sleep in working-age adults are lacking. One study in female office workers in the U.S. found that daytime MVPA was not associated with that night’s TST or SE [5], as in the present study, whereas another study of older women contrarily found that more MVPA was associated with less TST [34]. The latter study had no employment information and included women with an average age of 73 years. The lack of agreement between the previous studies may partly be attributable to neither a mutual control for sedentary and MVPA behaviors, although important confounders, nor simultaneous control of average levels of these behaviors. Moreover, these studies focused predominantly on women, whereas the present study included both sexes. Another study of overweight/obese adults also found no association between MVPA and the same night’s TST and SE [35]. The authors explained this result as being a consequence of the population having a higher body mass index and low levels of MVPA, as a minimum threshold of MVPA might be required to affect sleep. However, our sample reached recommended MVPA levels (150–300 min/week) [36] and had good fitness [30]. Other studies in persons with sleep disorders or of older age have reported positive effects of within-person physical activity on sleep and vice versa, although not all studies employed objective measurement tools [33,34,37,38,39]. For example, a study including adults aged 53–101 years old demonstrated the contrary—that more MVPA was associated with longer sleep times [38]. Thus, physical activity’s effects on sleep may be more effective in persons with sleep disturbances or in older adults.

Furthermore, MVPA did have an association with sleep when sex was taken into consideration as a modifier in the present study. More daytime MVPA was associated with less SE that same night only in men. Work schedules and family responsibilities are important factors that may influence sleep-wake cycles, among both men and women. However, in the present study women had longer sleep durations on average than men, with men’s mean sleep duration being just under the recommendations of seven or more hours per night [40]. Besides work-life balance, other factors such as timing of the exercise and proximity of the exercise to sleep time may have affected sleep quality. One study assessed the acute effects of daily aerobic exercise in young adult men over six nights [41]. They reported that circadian melatonin rhythm, nocturnal rectal temperature, sleep stages, and heart rate variability differed depending on the time of day the moderate intensity activity was performed. Thus, exercise performed earlier in the day may improve sleep quality by stimulating the sympathetic nervous system [41]. Furthermore, a study on working men’s perspectives about sleep health reported that men slept on average 6.4 h per night and normalized sleep deprivation in order to meet workplace ideologies [42].

In regard to the effects of daytime sedentary behavior on sleep, our findings were partially supported by a study including overweight/obese adults. In that study, Imes et al. found that more sedentary time was associated with less TST [35]. Contrarily, Mitchell et al. found no association between sedentary time and TST [5]. A study using Fitbit data also found a negative within-person association over six months, suggesting that more minutes spent sedentarily during the day was associated with less sleep at night [43]. Pettee Gabriel et al. reported that more sedentary time was associated with shorter sleep duration, defined as <7 h in older women in the U.S. [44]. However, these studies differed in their populations compared with the present study, with a predominant focus on women and older populations. Moreover, the association between sedentary time and TST was also more pronounced in those who were older in the present study, although still younger than the previously reported studies. Nevertheless, it is suggested that sleep duration [45] and the positive effects of exercise and sleep diminish with age [46]. In a meta-analysis by Ohayon et al. it was reported that TST decreased linearly with age, exhibiting a loss of 10 min per decade of age [45]. This may be a consequence of redistribution of time. Another explanation may be the negative effects of sedentary behavior on sleep, which may be crucial for an older population who are more likely to be sedentary [47]. Sedentary behavior and sleep disturbances are risk factors for early death [8,13], mental illness [48], and cognitive impairment [49,50]. For this reason, older office workers may be critical targets for interventions to help reduce sedentary time and improve sleep for continued good working ability and independence in daily living. Moreover, when examining the relationship between sedentary time and SE, Imes et al. also found no significant association [35], similar to our study, and Mitchell et al. [5].

Reciprocally, sleep’s association with physical activity and sedentary behavior the following day may suggest that this direction of the association is of greater importance among office workers than MVPA and sedentary time on sleep. A bi-directional relationship between TST and sedentary time was also found among overweight/obese adults in the study by Imes et al. [35]. They reported that 60 min additional TST was associated with 19.2 min less sedentary time the next day. However, they did not find an association between TST on MVPA, which was contrary to the present study. A study in female office workers in the U.S. [5] and older women [34] also did not find an association between TST and MVPA, whereas another study of older women found that long sleep (>9 h) was associated with less MVPA the following day [44]. The results from the latter study were similar to the current study, except we found this association to be only present on weekdays and not weekend days. The study of overweight/obese adults observed similar findings, such that more time spent sleeping, particularly on weekdays, was associated with less sedentary time the following day [35]. This may suggest that office workers are more likely to prioritize sleep over other activities on workdays. According to the results in the present study, long TST on weekdays may replace both sedentary and physical activity time in this population. This may seem optimistic considering that sedentary time is being reduced and sleep time extended. Nevertheless, too long a sleep, as aforementioned, can also have negative consequences for health and increase the risk of early death, just as sleep deprivation can, indicating a U-shaped relationship [13,14]. Therefore, ideally, it should be advised to replace prolonged sleep with physical activity rather than sedentary time. This can also be of benefit considering that more sedentary behavior corresponds to less TST, regardless of what day of the week it is, and the known benefits of increasing physical activity on health in general. In addition, prolonged sleep concomitantly with a reduction in physical activity and sedentary behavior may point to other underlying health conditions, such as mental illness [51] or cardiovascular conditions [14]. Differential effects from weekdays to weekend days may indicate a behavioral alteration with regard to the day of the week. As a result, interventions may consider focusing on improving behavioral changes on weekdays to improve health among office workers.

Furthermore, SE showed a relationship with MVPA on the following day, which was likewise observed in a study of older women [34], such that higher SE during the night was associated with more time in MVPA the following day. Therefore, having good sleep quality may affect physical activity the following day even after taking sedentary time into account, as demonstrated in the current study. Indeed, the present population may be considered to have good sleep quality according to recommendations for SE (>85%) [27]. It is suggested that SE should ideally lie between 85% and 90% [27], where the current population had a median SE of 91%. Too high sleep efficiency has been suggested to be associated with feelings of not being sufficiently rested, though observed in patients with insomnia [27]. However, the association between SE and the following day’s MVPA was no longer significant after controlling for stress in the additional analysis. The negative consequences of having SE levels above the recommended levels in the present population may also partly explain the positive association between SE and sedentary behavior. Nevertheless, it is more likely that these findings indicate a tendency towards being more efficient and concentrated at work the following day, resulting in more time spent seated during work tasks. This may be the case in this population, since a study of Swedish office workers reported on average that they favored sitting during worktime and perceived changing positions as cumbersome during the workday [52]. Thus, despite sleep efficiency’s potential advantageous effects on increasing work-related concentration, consideration should be given to enhancing physical activity breaks during the workday to substitute the sitting time for overall health benefits.

In regard to other effect modifiers, such as fitness, earlier studies have indicated a modifying effect on the association between sleep and physical activity [4]. However, we did not find this in the current study. This may be owing to our relatively healthy and active population. A previous study in older adults also exhibited no modifying effect of cognitive function [38]. Moreover, cognitive performance is also linked to mood and stress, which are plausible pathways by which sleep and physical activity may interact [37,53]; however, the associations did not differ much, in the current study, when self-perceived stress was controlled for. Therefore, more studies are needed to confirm these associations. Other potential physiological mechanisms by which day-to-day or regular physical activity affects sleep—although we were unable to test these in our study—are thermoregulation, endocrine function, alterations of the central and autonomic nervous system, and immune function [17].

### Strengths and Limitations

Strengths of our study include the use of valid device-based measures of sleep and physical activity behaviors over at least four days, and the use of sleep diaries to anchor times. In addition, the inclusion of temporal associations gives an added advantage as most studies have predominantly relied only on cross-sectional assessments at one time point. The simultaneous adjustment of MVPA and sedentary behavior in our models provides a more realistic image of these behaviors as they do not occur independently of each other. There may be a lack of generalizability to other populations that are not working, other companies, and to populations with underlying health issues. This may be evident in those who were excluded because of lacking accelerometer data. These persons were more likely to have lower scores on the executive function composite score. In addition, the current sample in the present study had, on average, a high education level. Furthermore, the companies included already had policies in place for their employees’ health, and in Sweden, most companies provide adjustable workstations and wellness benefits to their employees, which may have led to potential sampling bias. This needs to be taken into consideration in future investigations of office workers, as well as heterogeneity in job positions within workplaces. The current sample consisted only of office workers from two companies in Sweden, with limited information on specific job-related tasks. Thus, results may not be comparable to companies with more diverse job positions. In addition, this study focused on short-term associations, while the long-term effects may reveal differing results. Notwithstanding, average values for the different parameters were also controlled for in the models. In addition, there may be some residual confounding, as we were not able to control for all confounders.

## 5. Conclusions

In conclusion, sleep may be more strongly associated with physical activity and sedentary behavior on the following day for office workers than physical activity and sedentary behavior for that night’s sleep. Particularly, extended sleep time on weekdays may reduce sedentary and physical activity behaviors on the following day, suggesting that on workdays, office workers struggle to make time for both sleep and physical activity. Interventions with the purpose to alter sleep as well as sedentary and physical activity behaviors of office workers should therefore take into consideration the interrelatedness between these factors and the day of the week on which they are performed.

## Figures and Tables

**Table 1 ijerph-18-07999-t001:** Means and standard deviations of study participants’ characteristics.

	Total (*n* = 324)	Women (*n* = 219)	Men (*n* = 105)	*p*-Value
Age, years	42.4 (9.0)	42.0 (9.2)	43.3 (8.6)	0.22
Education, years	14.4 (2.3)	14.2 (2.3)	14.8 (2.3)	0.03
Body mass, kg	74.1 (13.6)	69.2 (12.3)	84.4 (10.2)	<0.001
Fitness, VO_2max_ mL·kg^−1^·min^−1^	40.0 (8.4)	37.8 (7.8)	44.7 (7.6)	<0.001
Stroop test time, sec	47.9 (9.0)	46.8 (9.2)	50.0 (8.2)	<0.001
TMT-B time, sec	52.4 (15.4)	52.1 (14.9)	53.1 (16.4)	0.57
Two-back, correct	72.1 (7.4)	71.3 (8.0)	73.8 (5.8)	<0.001
Stress, score	2.7 (1.2)	2.8 (1.2)	2.4 (1.1)	<0.001

TMT-B: Trail Making Test-B. Missing 3 in age, 2 in education and body mass, 4 in fitness, 2 in Stroop, 8 in TMT-B, 2 in two-back, and 22 in stress. Stress ranged from 1 (low stress) to 5 (high stress). *p*-value for differences between women and men.

**Table 2 ijerph-18-07999-t002:** Average or median proportion/minutes of wear time spent in physical activity and sedentary time, and sleep parameters.

	Weekday		Weekend Day		All Days	
	Women	Men	Women	Men	Women	Men
Mean SED (SD), %	61.2 (9.4) *	62.9 (8.9)	50.5 (12.0) *	53.9 (12.3)	58.5 (11.2) *	60.6 (10.7)
Median MVPA (IQR), %	6.0 (4.2) *	5.6 (4.4)	6.6 (6.6)	6.1 (5.3)	6.1 (4.9) *	5.7 (4.5)
Mean TST (SD), minutes	415.9 (60.2) *	397.9 (65.1)	439.4 (78.3) *	420.7 (66.9)	421.6 (65.4) *	403.8 (65.1)
Median SE (IQR), %	91.4 (6.8)	91.1 (7.1)	90.8 (7.0)	91.2 (8.7)	91.1 (6.7)	91.1 (7.4)

SED: sedentary behavior; MVPA: moderate-to-vigorous physical activity; TST: total sleep time; SE: sleep efficiency; SD: standard deviation; IQR: interquartile range. * Behaviors significantly different between men and women between weekdays and weekend days, and over all days (*p* < 0.05). Note: sedentary and moderate-to-vigorous physical activity time are percent of awake wear time.

**Table 3 ijerph-18-07999-t003:** Multilevel linear mixed models showing the bi-directional, day-to-day associations between physical activity behaviors (MVPA and sedentary) during the day and that night’s sleep parameters (total sleep time and sleep efficiency), and sleep parameters’ associations with physical activity behaviors during the following day (*n* = 324).

	Standardized β-Coefficient (95% Confidence Interval)	
	Model 1	Model 2
Daytime physical activity and sedentary behavior’s association with that night’s sleep parameters		
	Total sleep time	
Sedentary behavior	−0.13 (−0.17, −0.09) ***	−0.13 (−0.17, −0.09) ***
MVPA	−0.04 (−0.09, 0.004)	−0.03 (−0.07, 0.02)
	Sleep efficiency	
Sedentary behavior	0.03 (−0.01, 0.06)	0.03 (−0.001, 0.07)
MVPA	−0.01 (−0.04, 0.03)	0.002 (−0.03, 0.04)
Sleep parameters’ association with physical activity and sedentary behavior on the following day		
	Sedentary behavior	
Total sleep time	−0.13 (−0.17, −0.10) ***	−0.13 (−0.17, −0.09) ***
Sleep efficiency	0.04 (0.01, 0.06) *	0.04 (0.002, 0.07) *
	MVPA	
Total sleep time	−0.07 (−0.10, −0.03) ***	−0.06 (−0.10, −0.03) **
Sleep efficiency	0.04 (0.01, 0.08) *	0.04 (0.01, 0.08) *

Model 1: control for MVPA or sedentary behavior; Model 2: control for Model 1 and age, sex, fitness, and cognitive performance. MVPA: moderate-to-vigorous physical activity; TST: total sleep time; SE: sleep efficiency. *p*-value * < 0.05, ** < 0.01, *** < 0.001. Variables were z-score-transformed prior to model parameterization.

**Table 4 ijerph-18-07999-t004:** Multilevel linear mixed models stratified by weekday/weekend for associations between physical activity, sedentary time, and sleep parameters.

	Standardized β-Coefficient (95% Confidence Interval)			
	Weekday		Weekend Day	
	Model 1	Model 2	Model 1	Model 2
	Total sleep time		Total sleep time	
MVPA	−0.06 (−0.12, −0.002) *	−0.05 (−0.12, 0.01)	0.03 (−0.06, 0.12)	0.07 (−0.02, 0.17)
Sedentary behavior	−0.05 (−0.12, 0.03)	−0.08 (−0.18, 0.01)	−0.04 (−0.12, 0.04)	−0.08 (−0.18,0.02)
	Sleep efficiency		Sleep efficiency	
MVPA	−0.01 (−0.06, 0.05)	−0.02 (−0.09, 0.05)	−0.004 (−0.06, 0.05)	−0.001 (−0.07, 0.07)
Sedentary behavior	0.04 (−0.03, 0.11)	−0.05 (−0.13, 0.02)	0.03 (−0.04, 0.10)	−0.04 (−0.12,0.04)
	Sedentary behavior		Sedentary behavior	
Total sleep time	−0.06 (−0.10, −0.02) **	−0.06 (−0.10, −0.02) **	0.03 (−0.02, 0.09)	0.03 (−0.03, 0.09)
Sleep efficiency	0.02 (−0.01, 0.06)	0.03 (−0.03, 0.08)	0.02 (−0.02, 0.05)	0.03 (−0.03, 0.08)
	MVPA		MVPA	
Total sleep time	−0.07 (−0.11, −0.02) **	−0.07 (−0.11, −0.02) **	−0.02 (−0.08, 0.05)	−0.02 (−0.09, 0.05)
Sleep efficiency	0.05 (0.01, 0.10) *	0.04 (−0.03, 0.10)	0.06 (0.01, 0.10)*	0.03 (−0.04, 0.09)

Model 1: control for MVPA or sedentary behavior; Model 2: control for Model 1 and age, sex, fitness, and cognitive performance. MVPA: moderate-to-vigorous physical activity; TST, total sleep time. *p*-value * < 0 0.05, ** < 0 0.01. Variables were z-score-transformed prior to model parameterization.

## Data Availability

Data availability will be considered by the principal investigator (maria.ekblom@gih.se) based on reasonable request.

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
