# Peer review of "Bi-Directional, Day-to-Day Associations between Objectively-Measured Physical Activity, Sedentary Behavior, and Sleep among Office Workers"

_ijerph, 2021, doi:10.3390/ijerph18157999_

Round 1

Reviewer 1 Report

This is a very interesting manuscript that deals with an important issue of the temporal interplay between different movement behaviour. It asks whether more time spent sleeping has effect on the behaviour the following day. The manuscript is well written, the data recording and processing detailled adequately but the statistical analysis is very opaque. It is unclear how the exposure and outcome variables are entered in the model. Are these entered as daily summary? How are the difference between week day and weekend ascertain. Is this through a dummy variable? Before I can comment or review the results, which by the way seem interesting and seem to make sense, I need further explanation of the analytic techniques. The author should detail this in a more digestible way, may be using equations or graphics. They could use a supplementary material to really give the reader a full understanding and complete transparency on their modelling.  

Since this is an observational study, I would expect the author to follow STROBE guidelines and conduct and report sensitivity analyses.

A major limitation of the approach that the author took to investigating this questions and that is not reflected upon in the manuscript is that they assume that the impact of one behaviour on the other is on a one day timescale? Is this a reasonable assumption? This should be reflected in the title "day to day". 

I am also quite surprised to the lack of attention paid to the actual occupation of each participants. Surely, their seniority and type of job will strongly influence the relationships probed. 

Author Response

Thank you for your comments! Below you can find our response.

This is a very interesting manuscript that deals with an important issue of the temporal interplay between different movement behaviour. It asks whether more time spent sleeping has effect on the behaviour the following day. The manuscript is well written, the data recording and processing detailled adequately but the statistical analysis is very opaque. It is unclear how the exposure and outcome variables are entered in the model. Are these entered as daily summary? How are the difference between week day and weekend ascertain. Is this through a dummy variable? Before I can comment or review the results, which by the way seem interesting and seem to make sense, I need further explanation of the analytic techniques. The author should detail this in a more digestible way, may be using equations or graphics. They could use a supplementary material to really give the reader a full understanding and complete transparency on their modelling.  

RESPONSE:  We have now revised the statistical analysis section in the paper (Section 2.6) to add more detail, included a reference (#28 Dzierzewski et al.), and added the equations in the Supplementary Material 1.0 Statistical analysis.

Since this is an observational study, I would expect the author to follow STROBE guidelines and conduct and report sensitivity analyses.

RESPONSE: A sensitivity analysis was done by controlling for confounders, as seen in Model 2, in order to consider any potential confounding. This tested whether any of the associations differed as a function of different confounders. Additional analyses were performed to test whether self-perceived stress and educations were also confounders (see Supplementary Material Table 1), however there were several observations missing (7%) in the stress variable. In addition, additional logistic mixed-effects models were performed where sleep efficiency and MVPA were the outcomes, to test whether this changed the results when presented as dichotomous outcomes, as they were skewed and log transformed in the original analysis. The additional analysis was now presented in the statistical analysis section (section 2.6), the results (section 3.4), and in the Supplementary Material (Tables 2 and 3). The results were similar when stress and education were controlled for and when the logistic mixed-effects models were performed.

A major limitation of the approach that the author took to investigating this questions and that is not reflected upon in the manuscript is that they assume that the impact of one behaviour on the other is on a one day timescale? Is this a reasonable assumption? This should be reflected in the title "day to day". 

RESPONSE: Thank you for your comment. We have now revised the title to include “day-to-day” for clarity, and throughout the text.

We have also added a part in the introduction to emphasize the need to look at short-term associations and added some more references (line 45, 53-57).

I am also quite surprised to the lack of attention paid to the actual occupation of each participants. Surely, their seniority and type of job will strongly influence the relationships probed. 

RESPONSE: This population consists of only office workers from two offices in Sweden. Therefore, the occupational status is quite homogeneous. In addition, the sample on average had a high education. However, we did not have job-specific information in the data. We have added a sentence in the limitations section. However, the effects of diversities in job positions on physical activity, sedentary behavior, and sleep may manifest instead in perceived stress. Therefore, we have provided a supplementary analysis including self-perceived stress as a confounder in the models to see if the associations differ (Section 3.4; Supplementary Material Table 1). However, the results were quite similar to those reported in the main manuscript. This is reported in the results (section 3.4) and discussion (lines 390 and 408-409).

Reviewer 2 Report

The study examined the bi-directional, within-person associations between daytime PA, sedentary behavior, and sleep in office workers, and found that sleep had a more important effect on MVPA on the following day, than daytime MVPA on sleep, as well as the signifcant association between sleep and sedentary behavior. My concerns are listed below.

  1. My biggest concern would be: using convenience sampling, the representativeness of the sample may be problematic. Authors stated in Strenghts and Limitations that our participants are representative of a large proportion of the population, which is not true. Considering the possible significant differences in cognitive performance between analytical sample and participants with missed data, and the possible substantial differences in demographic characteristics between participants(1940 persons) and non-participants, authors should systematically discuss this issue.
  2. The study investigated the short-term effect of sleep on PA an sedentary behavior, and vice versa. So the longer-term effect was not taken into account, which may be significantly difference from the short-term effect. Therefore, I think this would be one of the limiations of the study. Also, it should be disccused in the Discussion section.
  3. The paper claimed to study a middle-aged population, yet in the "results" the age range was described as 24 to 66 years, which did not appear to be exactly middle-aged.
  4. Minor issue: The first appearance of the abbreviation (MLMs) requires the full name.

Author Response

Thank you for your comments! Below you can find our responses.

  1. My biggest concern would be: using convenience sampling, the representativeness of the sample may be problematic. Authors stated in Strenghts and Limitations that our participants are representative of a large proportion of the population, which is not true. Considering the possible significant differences in cognitive performance between analytical sample and participants with missed data, and the possible substantial differences in demographic characteristics between participants(1940 persons) and non-participants, authors should systematically discuss this issue.

RESPONSE: There may be an issue of potential sampling bias, thus this limitation has now been mentioned in the limitations section (section 4.1).

There may be a lack of generalizability to other populations that are not working, other companies, and to populations with underlying health issues. This may be noted by those who were excluded, due to not having accelerometer data, having lower scores on the executive function composite score. In addition, the companies included already had policies in place for their employee’s health, and in Sweden, most companies provide adjustable workstations and corporate wellness benefits to their employees, which may have led to potential sampling bias. In addition, the sample in the current study had a high education level. This needs to be taken into consideration in future investigations of office workers.

  1. The study investigated the short-term effect of sleep on PA an sedentary behavior, and vice versa. So the longer-term effect was not taken into account, which may be significantly difference from the short-term effect. Therefore, I think this would be one of the limiations of the study. Also, it should be disccused in the Discussion section.

RESPONSE: We have added a sentence now in the limitations (section 4.1, lines 432-433).

In addition, this study focused on the acute associations between these behaviors, whereas the long-term effects may reveal differing results. However, the weekly averages of each of the behaviors were also controlled for in the models. We have also clarified in the methods that we control for weekly averages and have emphasized in the introduction that this study focuses on short-term associations.

  1. The paper claimed to study a middle-aged population, yet in the "results" the age range was described as 24 to 66 years, which did not appear to be exactly middle-aged.

RESPONSE: We have now changed “middle-age” to “working-age” throughout the manuscript.

  1. Minor issue: The first appearance of the abbreviation (MLMs) requires the full name.

RESPONSE: This has now been fixed in section 2.6, line 160.

Reviewer 3 Report

Paper title: Bi-directional, within-person associations between objectively-2 measured physical activity, sedentary behavior, and sleep 3 among office workers

GENERAL COMMENTS

This study investigated the bi-directional, within-person associations between daytime physical activity and sedentary behavior, and nocturnal sleep.

The study is very interesting, presents important data and a considerable number of participants. This is valued, given the effort required from the researchers to be able to objectively measure the level of physical activity through accelerometry.

However, there are two elements that concern me. The first is about the intensity in Hz and the Epochs used during sleep time. This could mean an important methodological element to consider and that could lead to misinterpretation. Literature should be reviewed in this regard. Second, it is important to be able to restructure the discussion, since various comparisons are made with groups of older adults, which are not related to the specific group of the present study.

SPECIFIC COMMENTS

Introduction

Line 62-63: “Should be” has been used twice in the same sentence. Please correct.

Lines 66-67: the sentence “In addition, cognitive performance may moderate the association [16], as suggested in 66 older adults, but requires testing in a middle-age population”, is not related with field of study and same with before and after paragraphs. Better delete.

Line 71: TST, not has been explained before. Is necessary include this topic in introduction.

Material and methods

Lines 89-90: in these lines, speak about a “monitor”, but this has not been explained previously.

Lines 98-100: Were frequency (Hz) and epochs same in both, walking hours and sleep time?. Please, review and add the missing data.

Line 117: Please cite the formula presented.

Line 140: What about MLMs? Please explain.

Results

I do not see it necessary to include in Table 1, information such as BMI or Vo2max, which is not explained in discussion and are not outcomes of the study.

Tables 3 and 4: For a better understanding, only use two decimals’ values in Coef. Stand. Beta, and CI. Is not easy see the table with a lot of numbers.

Discussion

Line 234-235: According the sentence: “was observed to have a more important effect on MVPA on the following day, than daytime MVPA on that night’s sleep”. But, where we can observe the effect on MVPA on the following day? Please explain better.

Throughout the discussion, references made in older adults are mentioned. I do not see it appropriate to compare groups of older people with this study. There is not much evidence about it, but not different groups. I suggest removing those references and those paragraphs from the discussion. Also, look for new information available.

Conclusion

One cannot speak of a "predictor" if a prediction statistic has not been done. In this case they are just "associations".

Author Response

Thank you for your comments! Below you can find our response!

 SPECIFIC COMMENTS

 Introduction 

Line 62-63: “Should be” has been used twice in the same sentence. Please correct.

RESPONSE: This has been revised. 

Lines 66-67: the sentence “In addition, cognitive performance may moderate the association [16], as suggested in 66 older adults, but requires testing in a middle-age population”, is not related with field of study and same with before and after paragraphs. Better delete.

RESPONSE: We have revised the text in the introduction (lines 73-78).

 Line 71: TST, not has been explained before. Is necessary include this topic in introduction.

RESPONSE: Sleep duration was mentioned in the introduction, which is synonymous to total sleep time. We have changed it to TST to increase clarity (line 49).

 Material and methods

 Lines 89-90: in these lines, speak about a “monitor”, but this has not been explained previously.

RESPONSE: The monitor is explained earlier in line 98. We have now added “activity” in front of “monitor” on line 104.

Lines 98-100: Were frequency (Hz) and epochs same in both, walking hours and sleep time?. Please, review and add the missing data.

RESPONSE: Frequency and epochs were the same in both waking and sleep time. We used the Actigraph GT3X, which only allows for a sampling frequency of 30Hz. There is no consensus currently on how the data are to be processed; however, we followed the some of the guidelines reported in Migueles et al. 2017. The sampling frequency produced counts that need to be summed into specific time intervals or epochs, and this is usually done by applying specific intensity cut-points and algorithms, as we have explained. We have now revised to explain that the same frequency and epochs were used for waking and sleep times in section 2.2 and the reference added (Migueles et al. 2017) (lines 113-115).

 Line 117: Please cite the formula presented.

RESPONSE: We have now added in a reference for the sleep efficiency equation (line 132).

 Line 140: What about MLMs? Please explain.

RESPONSE: This has now been explained in the statistical analysis section (2.6) and in the supplementary material section 1.0.

 Results

 I do not see it necessary to include in Table 1, information such as BMI or Vo2max, which is not explained in discussion and are not outcomes of the study.

RESPONSE: VO2max, a measure of fitness, was included as a confounder and tested as a modifier in the study and the results explained in the discussion (see line 402). We do not report BMI, but rather body mass. It was not included in the models due to multicollinearity with VO2 max (correlation of 0.3). However, we kept it in Table 1 to add to the demographic description of the population.

 Tables 3 and 4: For a better understanding, only use two decimals’ values in Coef. Stand. Beta, and CI. Is not easy see the table with a lot of numbers.

RESPONSE: We have now reduced the values to 2 decimal points.

 Discussion

 Line 234-235: According the sentence: “was observed to have a more important effect on MVPA on the following day, than daytime MVPA on that night’s sleep”. But, where we can observe the effect on MVPA on the following day? Please explain better.

RESPONSE: This summary refers to the associations between TST and SE at night on MVPA the following day. These results are shown in Table 3, bottom half, “Sleep parameters’ associations with physical activity and sedentary behavior on the following day”. No association was found between MVPA during the day and sleep at night (in the top half of the table 3), there was with TST and SE with MVPA (bottom half of the table). We have revised the sentence in the Discussion to increase clarity (lines 277-280).

Throughout the discussion, references made in older adults are mentioned. I do not see it appropriate to compare groups of older people with this study. There is not much evidence about it, but not different groups. I suggest removing those references and those paragraphs from the discussion. Also, look for new information available.

RESPONSE: We believe it is still useful to compare to other populations, including older adults and patients, etc. Studies are limited that have investigated the day-to-day associations in working adults, therefore it is worth mentioning studies that have a similar design, although with slightly different age groups. However, we have compared to other studies in the discussion and not only older adults. If studies including older adults were mentioned it was clearly stated for the reader to consider a potential age discrepancy with the current study. Nevertheless, the definition of older adult does vary and the different studies with older adults have varying population ages. Our study also included persons up to 66 years old. In addition, comparing to older adults may be relevant considering that older adults also spend a considerable portion of their day sedentary. Furthermore, an indication of an association in an older population can provide some preclinical information to compare to a working aged population. We have, however, taken your comment into consideration, and have made some revisions by minimizing information from studies including older adults (lines 380-382 were removed some results).

 Conclusion

 One cannot speak of a "predictor" if a prediction statistic has not been done. In this case they are just "associations".

RESPONSE: We have now made the changes in the text accordingly.

Round 2

Reviewer 1 Report

I am satisfied with the revision made

Author Response

Thank you for your review!

Reviewer 2 Report

Last question: "However, the short-term associations remain unclear." This sentence seems come out of nowhere. Please improve the writing flow.

Author Response

Thank you for your review! We have now revised the sentence (line 45).

Reviewer 3 Report

Thank you for the review and changes done.

Regards,

Author Response

Thank you for your review!